# Study on the Skin Hydration and Trans Epidermal Water Loss of Aloe Viscose Seamless Knitted Fabric for Autumn and Winter

**DOI:** 10.3390/ma16010212

**Published:** 2022-12-26

**Authors:** Xiaokang Wang, Zimin Jin, Liumeng Mao, Lexi Tu, Yuqiang Sun, Jianwei Tao

**Affiliations:** 1College of Textile Science and Engineering, Zhejiang Sci-Tech University, Hangzhou 310018, China; 2College of Life Sciences and Medicine, Zhejiang Sci-Tech University, Hangzhou 310018, China; 3Zhejiang Bangjie Holding Group Co., Ltd., Yiwu 322000, China

**Keywords:** aloe viscose fiber, seamless knitted fabric, moisturizing property

## Abstract

To explore the skin moisturizing performance of aloe viscose fiber seamless knitted fabric, this experiment takes the different yarn-blending ratios of aloe viscose fiber and viscose fiber, as well as three different tissue structures as factors, establishes a sample scheme according to full factor experimental tests on skin hydration and trans epidermal water loss (TEWL) after the sample fabric had been wrapped around the skin, and uses two-way and one-way ANOVA in SPSS and the Duncan multiple comparison method. The test data were analyzed to study the influence of different materials and the structure of the veil on the moisture retention of the fabric. The results show that the sample scheme with the largest change rate of skin hydration is when the raw material of the veil is aloe viscose/viscose 100/0 yarn, and the structure is 1 + 3 simulated rib. The sample scheme with the minimum change rate of TEWL is when aloe viscose/viscose 75/25 yarn is used as the raw material of the veil, and the structure is 1 + 1 simulated rib, which provides a theoretical basis for the research and development of moisturizing knitted fabric.

## 1. Introduction

Bayer first proposed the concept of skin care finishing, and it is mainly a reference for cosmetics and skin care products. The so-called skin care textile refers to the application of some natural raw material extracts with skin care functions, such as ingredients with moisturizing, conditioning and other special effects, to textiles through post finishing. In this way, the functions can be endowed to meet people’s needs. In the late 1980s, in Japan, some effective ingredients were added to cosmetics and active substances added into textiles for the first time. In 2006, the British Bureau of Textile and Clothing Industry Standards officially defined the term skin care textile. It belongs to the category of textile consumer goods, and the active substances in the fabrics can be continuously released when wearing them, thus playing a role in skin care. Wearing it can keep the human body in a comfortable state [1,2,3,4].

Most textiles directly contact the skin, which is very fragile and sensitive, especially in autumn and winter, when the weather is dry, the skin easily becomes rough, and the nerve endings are more vulnerable to external stimulation due to the peeling off of the epidermis, resulting in skin itching, dryness and other uncomfortable symptoms. In view of this situation, it is particularly important to develop a fabric with good moisture retention [5].

Aloe fiber is a new type of skincare and healthcare fiber product. The active ingredients, such as aloe polysaccharide and amino acids in the aloe fabric, constitute a moisturizing factor, which can protect, moisturize and regulate the humidity of the skin [6]. Aloe has a good moisturizing effect on human skin. The application of aloe to textile fabrics gives the fabric a certain moisturizing effect on the skin. However, most of the aloe textiles on the market use post finishing. The product has poor water-washing resistance, and the moisturizing effect lacks long-term effectiveness. The serviceability of the product is often ignored [7]. Therefore, this paper uses microencapsulation to fix aloe on viscose fiber to obtain aloe viscose yarn [8], combines functional fiber and seamless knitting to design seamless knitting samples with different yarn-blending ratios and different knitting structures [9], and conducts moisture-retention performance tests on these samples to analyze the relationship between different yarn-blending ratios and different knitting structures and moisture retention performance.

This research is intended to obtain better process parameters, so as to design knitted clothing with good moisturizing effects, which provides a reference for the production of moisturizing textiles. For people who need skin moisturizing in autumn and winter, the research results can lay a theoretical foundation for their dress, and then provide more choices for them to buy, which meets the functional and personalized needs of consumers.

## 2. Materials and Methods

### 2.1. Materials

#### 2.1.1. Selection of Yarn Scheme

The raw materials of the veil were 11.81 tex (50s) aloe viscose yarn, aloe viscose/viscose blended yarn developed by Shandong Hengfeng New Type Yarn and Fabric Innovation Center Co., Ltd. (Dezhou, China) and 11.81 tex (50s) viscose yarn developed by Shandong Dezhou Huayuan Ecological Science and Technology Co., Ltd. (Dezhou, China). The active ingredients of aloe in the aloe viscose yarn were wrapped in micro particle capsules by microencapsulation, and the micro particle capsules were added with additives to obtain the aloe spinning solution. Then, it was mixed with viscose stock solution and spun to obtain aloe viscose fiber [10]. The principle was to completely wrap the target with a continuous film of the polymer compound, without any damage to the original chemical properties of the target. Then, gradually, the function of the target was displayed externally again through slow release, or the shielding the capsule wall played a role in protecting the target, improving the stability of the product and preventing interference between the various components. In this method, the active ingredients were slowly released to ensure the long-term effect [11,12].

The lining yarn was made of 2.22 tex/3.33 tex (20D/30D) nylon/spandex covered yarn provided by Yiwu Huading Nylon Co., Ltd. (Yiwu, China) to improve the elasticity of the designed knitted fabric.

#### 2.1.2. Fabric Structure Design

The tissue structure plays an important role in the appearance and wearing performance of knitted fabrics. In addition, to meet the needs of each clothing area, different parts of the clothing also need different tissue structures [13]. Therefore, in combination with the actual production requirements and winter clothing requirements, three kinds of structures were designed, namely, weft plain, 1 + 1 simulated rib, and 1 + 3 simulated rib. The structure chart is shown in Figure 1.

Weft Plain;

The weft plain knitted fabric is a single-sided weft knitted fabric. Its front and back sides are different. The front side is braided and the back side is corrugated [14]. Compared with other knitted fabric structures, the structure of weft flat needle is relatively simple, and the amount of yarn used for knitting on the machine is small, which is mainly used to manufacture large body sections of clothing fabrics.

2.Simulated rib;

1 + 1 simulated rib fabric and 1 + 3 simulated rib fabric belong to the category of double-sided weft knitted fabric. Although the simulated rib weave has the same appearance effect as rib weave, its elasticity and ductility are worse than rib weave, and it is often used to make necklines, sleeves, trouser legs, and other parts of clothing. When the simulated rib fabric is woven, because the fabric has long floating lines, the loop is stretched during weaving. When there are many continuous and non-woven rows, the mesh effect is easy to form on the surface of the fabric, showing a relatively loose state [15].

#### 2.1.3. Establishment of Sample Scheme

In this paper, the raw materials and structure of the veil were taken as factors, among which the blending ratio of aloe viscose/viscose yarn as the raw materials of the veil had 5 levels, namely 0/100, 25/75, 50/50, 75/25, 100/0, and the structure had 3 levels, namely, weft plain, 1 + 1 simulated rib, 1 + 3 simulated rib. The full-factor test design method [16] was adopted, and the specific fabric sample scheme is shown in Table 1.

### 2.2. Methods

#### 2.2.1. Test of Skin Hydration

Skin hydration plays an important role in regulating the skin and even the whole body, which is closely related to the skin’s soft, smooth, elastic, and other visual perceptions [17]. By increasing skin hydration, we can clearly know the biological state of the body, delay skin aging, and prevent some skin diseases [18].

Experimental Equipment;

The model of skin hydration detector used in the experiment was Probe Corneometer^®^ CM825, as shown in Figure 2b, produced by CK Company in Germany. The working principle of the instrument is based on the fact that the dielectric constant (<7) of water and other substances changes considerably. According to a difference in skin moisture content, the capacitor will change with a corresponding change of skin capacitance when measuring the skin, and the skin electrical capacity thus obtained is within the appropriate value range. The skin moisture content obtained in this way is expressed in MMA (0–150) [19]. When the tested area and the test probe are reasonably and effectively bonded, there is basically no small current passing through the tested area, and the test results are relatively accurate [20].

2.Experimental Materials and Test Conditions;

When designing the size of the sample specification, it was necessary to fully consider the size of the subject’s body and arm circumference, so that there was no pressure applied when the sample was wrapped around the forearm of the subject during the test. The specific size of the sample was set as follows: the width was 15 cm, the length was 20~23 cm, and 2 cm wide velcro was set on both sides to ensure that the sample completely wrapped the forearm and the sample size could be properly adjusted at the same time. The sample was wrapped against the skin and fixed gently; no pressure is required for wrapping and removal to avoid friction and to simulate natural static state. After coating the sample, all subjects were placed under a condition of 20 ± 2 °C and 40~60% relative humidity for the moisture content test of the skin cuticle. The number of people in this test was set at 15, all of whom were between 22 and 25 years old, without any history of skin and systemic diseases [21,22].

3.Experimental Steps.

Subjects could not use other skin care products and drugs at the test site 2 to 3 days before the test, and could not touch water 1 to 2 h before the test. Before the test, the forearm of the subject was wiped with tissue paper, and then two test areas were marked on the forearm of the subject with an area of 3 cm × 3 cm, with an interval of more than 1 cm between the two test areas. I was the test area of the covered sample part, and II was the test area of the uncovered sample part, as shown in Figure 3.

Subjects sat in a room that meets the test conditions for 20 min, and no water or drink was allowed during the test. The forearm was exposed so that the arm was naturally extended and in a relaxed state. When testing skin hydration, the probe was 90° to the forearm of the body, as shown in Figure 4. Skin hydration in area I and area II was tested, respectively, as the basic values $T_{1}$ and $T_{2}$. Each part was tested five times, and the average value taken. Then, the sample in area I was coated as shown in Figure 5. After 2 h, the two parts were tested again. The test result of area I of the covered sample was $T_{3}$, and that of area II of the uncovered sample was $T_{4}$, as the blank control group.

Skin hydration in area I of each sample covering the sample was calculated according to Formula (1), and the average value of each subject was taken as the test result of the change rate of skin hydration in area I of each sample covering the sample.
(1)$$R_{1}=\frac{T_{3}-T_{1}}{T_{1}}$$
among them: $R_{1}$ is the change rate of skin hydration covering area I of the sample site (%); $T_{3}$ is in area I after 2 h of the test (%); $T_{1}$ is in area I before the test (%).

Skin hydration in the uncovered sample area II of each sample was calculated according to Formula (2), and the average value of each subject was taken as the test result of the change rate of skin hydration in the uncovered sample area II of each sample.
(2)$$R_{2}=\frac{T_{4}-T_{2}}{T_{2}}$$
among them: $R_{2}$ is the change rate of skin hydration in area II of the uncovered sample part (%); $T_{4}$ is in area II after 2 h of the test (%); $T_{2}$ is in area II before the test (%).

#### 2.2.2. Test of Trans Epidermal Water Loss

Trans epidermal water loss (TEWL), which reflects the evaporation of water from the skin surface, is one of the main indicators to evaluate the skin barrier function [23]. Although the water loss rate through the epidermis cannot accurately and intuitively reflect the total water content of the stratum corneum, it can reflect the water loss of the stratum corneum. The higher the water loss rate through the epidermis, the more water will be lost through the skin per unit time, the weaker the natural barrier function of the stratum corneum, and the weaker the self-healing ability of the skin [24].

Experimental Equipment;

The model of TEWL detector used in the experiment was the Probe Tewameter^@^ TM Hex, as shown in Figure 2c, produced by CK Company in Germany. The principle of the instrument is based on Fick’s diffusion law—dm/dt = D·A·dp/dx. In this law, A is the area (m^2^); D is the diffusion constant 0.0877 g/m·g·mmHg; M is the diffusion amount of water (g); T is the time (h); P is the steam pressure (mmHg); and X is the distance of the measuring point from the skin surface (m).

The probe is mainly composed of two groups of temperature sensors and humidity sensors. The shape and size of the sensor can effectively reduce the interference and impact of indoor air circulation on the measurement results. By measuring the difference between the skin moisture content at the starting point and the ending point for a period of time, it can accurately reflect the speed of skin surface moisture loss [25].

2.Experimental Test Conditions;

The specs of the samples used were consistent with skin hydration. After the subjects covered the samples, they were placed under the conditions of 20 ± 2 °C and 40~60% relative humidity for TEWL. The age of the subjects was between 22 and 25 years old, and the number of subjects tested was 15, without any history of skin and systemic diseases.

3.Experimental Steps.

After skin hydration was tested, TEWL was tested immediately. During the test, the probe and forearm were naturally fitted at 90°, as shown in Figure 5. Each part was tested five times, and the average value was taken. After clicking the start button, the instrument will automatically collect TEWL values per second, and after 30 s, the instrument will automatically stop testing [26].

During the test, it was necessary to avoid breathing and air flowing on the probe, which may affect the results. Using the probe, the trans epidermal water loss rate of area I and area II, respectively, were tested as the basic values $V_{1}$ and $V_{2}$. After 2 h of testing, the two parts were tested again. The test result of area I of the covered sample is $V_{3}$, and that of area II of the uncovered sample is $V_{4}$, as the blank control group.

TEWL of each sample in area I where the sample was covered was calculated according to Formula (3), and the average value of each subject was taken as the test result of the change rate of TEWL of each sample in area I where the sample was covered.
(3)$$R_{1}=\frac{V_{3}-V_{1}}{V_{1}}$$
among them: $R_{1}$ is the change rate of TEWL in area I covering the sample site (%); $V_{3}$ is in area I 2 h after the test, g/(cm·h); $V_{1}$ is in area I before the test, g/(cm·h).

TEWL of the uncovered sample area II of each sample was calculated according to Formula (4), and the average value of each subject was taken as the test result of the change rate of TEWL of the uncovered sample area II.
(4)$$R_{2}=\frac{V_{4}-V_{2}}{V_{2}}$$
among them: $R_{2}$ is the change rate of TEWL in area II of the uncovered sample site (%); $V_{4}$ is in area II 2 h after the test, g/(cm·h); $V_{2}$ is in area II before the test, g/(cm·h).

## 3. Results

### 3.1. Research on Fabric Structure Parameters and Skin Hydration

Table 2 shows the results of the average skin hydration change rate of 15 subjects corresponding to each sample. According to the table, with sample #12, the skin moisture content changes greatly, and the raw material is aloe viscose/viscose 100/0 yarn with a 1 + 3 simulated rib. With sample #13, the skin moisture content changes little, and the raw material is aloe viscose/viscose 0/100 yarn with a weft plain. The ranking of the advantages and disadvantages of each sample on the change of skin moisture content is #12 > #8 > #9 > #11 > #4 > #6 > #3 > #10 > #7 > #5 > #2 > #15 > #14 > #13. In addition, the skin hydration of the uncovered sample parts decreases, and the covered is higher than the uncovered. This is because the test area is exposed and there is no shelter from clothing, and the skin moisture is constantly lost, while the skin hydration of the covered sample parts increases, which shows that clothing has a certain effect on skin moisture.

Through SPSS two-way analysis of variance [27], the relationship between raw materials, tissue structure and skin hydration has been explored. The results of the two-way variance analysis of the moisture content of the cuticle are shown in Table 3. The veil material has a significant difference (*p* < 0.05), the different structures have no significant difference (*p* > 0.05), the interaction between veil material and structure is not significant (*p* > 0.05), and the data are in accordance with a normal distribution.

With the veil raw material as a factor and the change rate of cuticle water content as a dependent variable, Duncan multiple comparison of the differences between the levels of the veil raw material was conducted with single factor ANOVA. The results are shown in Table 4. The data in the table were analyzed and studied, and the conclusions are as follows.

From the raw materials of the veil, aloe viscose fiber/viscose fiber 25/75, 50/50 and 75/25 are in the same subset, indicating that there is no difference between levels. Aloe viscose/viscose 0/100 and 100/0, respectively, account for one subset, among which 100/0 has the best moisturizing performance. With the increase in the content of aloe viscose fiber in the yarn, the skin moisture content increases, which is because aloe contains a lot of polysaccharides, amino acids and other substances. These are important components of moisturizing factors. In the microstructure, many hydroxyl groups in the polysaccharide will form hydrogen bonds with water and cross link with each other to form a network structure, so as to achieve the moisturizing effect. When the content of aloe increases, the moisturizing effect of the fabric on the skin is gradually improved. Although viscose fabric has good moisture absorption, it has no effect on skin moisture retention. Therefore, in terms of the use of raw materials, the influence of the fabric on the moisturizing performance of human skin is ranked from large to small as follows: aloe viscose/viscose fiber fabric (100:0 > 75:25 > 50:50 > 25:75 > 0:100).

### 3.2. Research on Fabric Structure Parameters and TEWL

Table 5 shows the results of the average rate of change of TEWL of 15 subjects corresponding to each sample. According to the table, with sample #7, the rate of TEWL changes little, and the raw material is aloe viscose/viscose 75/25 with a 1 + 1 simulated rib. With sample #4, the rate of TEWL changes greatly, and the raw material of the veil is aloe viscose/viscose 100/0, with a weft plain.

The change degree of the rate of TEWL in the coated area is not as large as the uncoated one. This is because the uncovered area is greatly affected by the temperature, humidity and air flow of the environment, showing a large fluctuation. After covering, although the effect of each sample is different, the floating degree is not as large as the former.

In addition, the change rate of TEWL of each covered area is higher than the uncovered, meaning that covering sample can reduce the rate of skin water loss to a certain extent, but the effect is not obvious. This is because TEWL is used to quantify the effectiveness of the cuticle as a barrier to prevent water loss, which can measure the water loss from the body without sweat. However, in a short period of time, the coverage of the aloe knitted fabric cannot actually repair the skin barrier.

Two factor ANOVA of change rate of TEWL of uncovered and the covered areas is shown in Table 6 and Table 7. From the table, the results show that there is no significant difference (*p* > 0.05) in the water loss of veil materials and structures, and the interaction between veil materials and structures is not significant (*p* > 0.05). Therefore, different veil materials and tissue structures have no obvious effect on TEWL.

## 4. Conclusions

Based on the factors of yarn-blending ratios of aloe viscose fiber and viscose fiber, as well as three different tissue structures, this experiment established a sample scheme according to a full-factor experimental design method to test the skin hydration and trans epidermal water loss after a sample fabric was wrapped around the skin. The two-way and one-way ANOVA in SPSS and Duncan multiple comparison method are used to analyze the data to study the influence of different yarn materials and tissue structures on the moisture retention of fabrics. The conclusions are as follows:In the moisture content test of skin, the influence of veil materials is significant, while the influence of tissue structure is not significant. The order of each sample is #12 > #8 > #9 > #11 > #4 > #6 > #3 > #10 > #7 > #5 > #2 > #15 > #14 > #13. In terms of the raw materials of the veil, aloe fabric is superior to viscose fabric. With the increase in the blending ratio of aloe viscose fiber in the yarn, the change rate of the moisture content of the cuticle increases. The sample scheme with a large change rate of the moisture content is that of the aloe viscose/viscose 100/0 yarn with a 1 + 3 simulated rib, which is sample #12;In the test of TEWL, the influence of veil raw materials and tissue structure is not significant. The sample scheme with a relatively small quantitative rate of TEWL is that with aloe viscose/viscose 75/25 yarn with a 1 + 1 simulated rib, which is sample #7.

In future research, it can be considered to set up a test experiment for the long-term wearing of fabrics, and add a subjective feeling questionnaire for subjects after it, which can completely evaluate textiles with moisturizing function and made into ready-made clothes for practical production.

## Figures and Tables

**Figure 1 materials-16-00212-f001:**
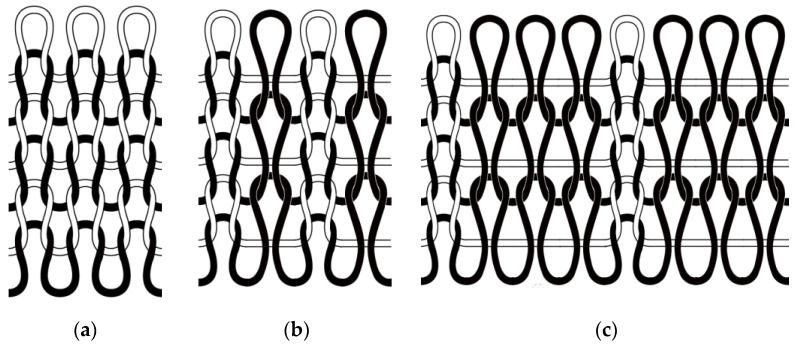
(**a**) The weft plain; (**b**) the 1 + 1 simulated rib; (**c**) the 1 + 3 simulated rib.

**Figure 2 materials-16-00212-f002:**
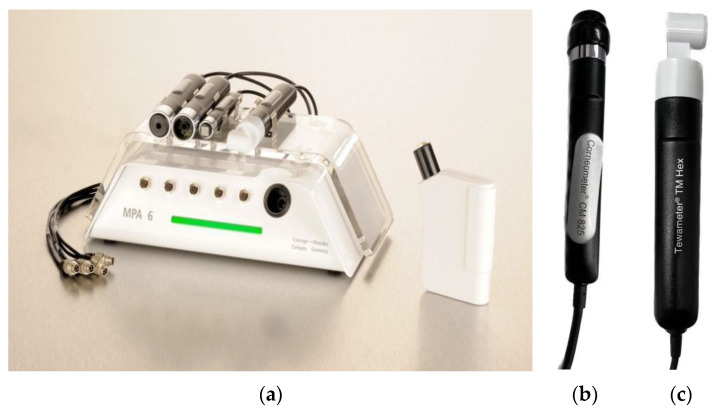
(**a**) Multifunctional skin tester MPA6; (**b**) Corneometer^®^ CM825; (**c**) Tewameter@ TM Hex.

**Figure 3 materials-16-00212-f003:**
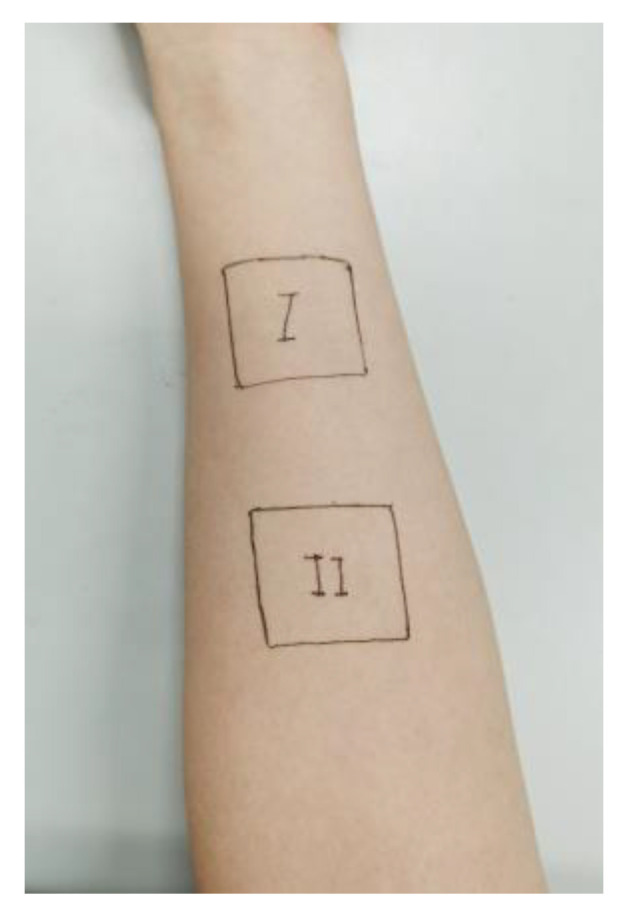
Marking of test area.

**Figure 4 materials-16-00212-f004:**
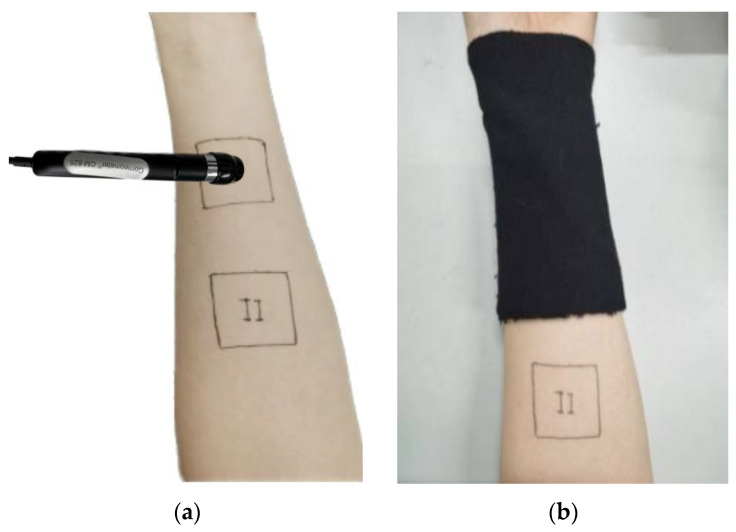
(**a**) Test of skin hydration by Corneometer^®^ CM825; (**b**) sample covering.

**Figure 5 materials-16-00212-f005:**
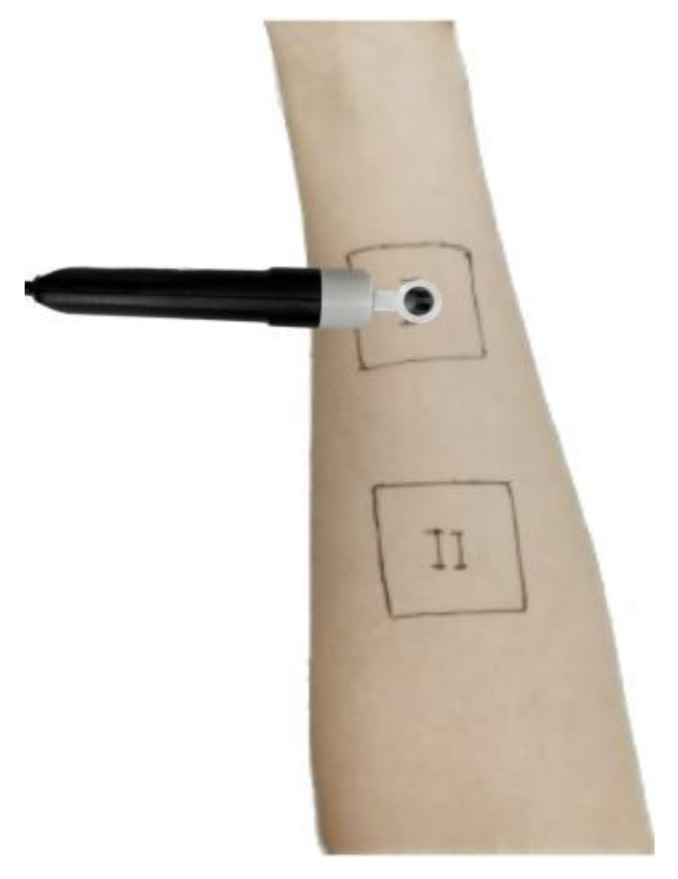
Test of TEWL by Tewameter@ TM Hex.

**Table 1 materials-16-00212-t001:** Veil materials and specifications.

Fabric Number	Veil Type	Fabric Weave
#1	Aloe viscose/viscose 25/75	weft plain
#2	Aloe viscose/viscose 50/50	weft plain
#3	Aloe viscose/viscose 75/25	weft plain
#4	Aloe viscose/viscose 100/0	weft plain
#5	Aloe viscose/viscose 25/75	1 + 1 simulated rib
#6	Aloe viscose/viscose 50/50	1 + 1 simulated rib
#7	Aloe viscose/viscose 75/25	1 + 1 simulated rib
#8	Aloe viscose/viscose 100/0	1 + 1 simulated rib
#9	Aloe viscose/viscose 25/75	1 + 3 simulated rib
#10	Aloe viscose/viscose 50/50	1 + 3 simulated rib
#11	Aloe viscose/viscose 75/25	1 + 3 simulated rib
#12	Aloe viscose/viscose 100/0	1 + 3 simulated rib
#13	Aloe viscose/viscose 0/100	weft plain
#14	Aloe viscose/viscose 0/100	1 + 1 simulated rib
#15	Aloe viscose/viscose 0/100	1 + 3 simulated rib

**Table 2 materials-16-00212-t002:** Change rate of skin hydration.

Sample Number	Change Rate of Skin Hydration of Covered/(%)	Change Rate of Skin Hydration of Uncovered/(%)
#1	3.07%	−4.17%
#2	4.06%	−2.36%
#3	4.96%	−1.43%
#4	5.20%	−0.42%
#5	4.55%	−1.26%
#6	5.15%	−0.31%
#7	4.63%	−0.79%
#8	6.50%	−1.02%
#9	5.88%	−0.55%
#10	4.90%	−1.74%
#11	5.49%	−1.23%
#12	8.03%	−1.89%
#13	0.11%	−2.55%
#14	0.47%	−1.73%
#15	1.19%	−0.60%

**Table 3 materials-16-00212-t003:** Two-way ANOVA of the change rate of skin hydration of covered areas.

Source	df	Mean Square	F	Sig.	Partial ETA Square
Raw material	4	154.43	5.50	0.00	0.11
Structure	2	44.29	1.58	0.21	0.02

**Table 4 materials-16-00212-t004:** Duncan multiple comparison table of change rate of skin hydration of different veil materials.

Subset of Alpha = 0.05			
Raw material	1	2	3
Viscose	1.19%		
25/75		4.19%	
50/50		4.58%	
75/25		5.49%	
100/0			6.47%

**Table 5 materials-16-00212-t005:** Change rate of TEWL.

Sample Number	Change Rate of TEWL of Covered Area/(%)	Change Rate of TEWL of Uncovered Area/(%)
#1	−4.48%	−12.55%
#2	−3.11%	−12.00%
#3	−2.15%	−12.12%
#4	−14.44%	−5.52%
#5	−2.83%	−8.69%
#6	−6.65%	−13.98%
#7	−1.56%	−10.63%
#8	−2.50%	−0.79%
#9	−3.34%	−5.59%
#10	−1.95%	−0.40%
#11	−6.17%	−11.20%
#12	−1.65%	−1.79%
#13	−2.13%	−2.29%
#14	−6.37%	−10.76%
#15	−4.69%	−17.39%

**Table 6 materials-16-00212-t006:** Two-way ANOVA of the change rate of TEWL of uncovered areas.

Source	df	Mean Square	F	Sig.
Raw material	4	113.45	0.25	0.92
Structure	2	469.75	0.98	0.38

**Table 7 materials-16-00212-t007:** Two-way ANOVA of the change rate of TEWL of covered areas.

Source	df	Mean Square	F	Sig.
Raw material	4	141.98	0.29	0.88
Structure	2	453.54	0.94	0.39

## Data Availability

All data can be found within the manuscript.

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
