# Peer review of "Study on the Skin Hydration and Trans Epidermal Water Loss of Aloe Viscose Seamless Knitted Fabric for Autumn and Winter"

_materials, 2022, doi:10.3390/ma16010212_

Round 1
Reviewer 1 Report
The article under review is devoted to the exploration of the skin moisturizing performance of aloe viscose fiber seamless knitted fabric. For this goal, authors estimated moisture content of the skin cuticle and water loss through the epidermis on 15 different samples. Each specimen was differ by tissue structure design and aloe viscose content. This article can be accepted for publication after major revisions:
1) introduction is very short and must be extended
2) discussion and conclusion sections must be separated
3) mistypes and mistakes should be corrected
4) conclusions need to contain workflow of the next investigations
5) improve plots from visual style point of view
6) correct the number of decimal places in tables
7) what is the goal of your study? why did you choose such materials? explain answers and insert them into the last paragraph on introduction section
Author Response
please check

Reviewer 2 Report
Dear authors,
the paper entitled Study on the moisturizing property of aloe viscose seamless 2 knitwear for autumn and winter
is very interesting as an idea and possible practical solution, but after reading the whole text and results, it seems to me that the title is not so appropriate for the paper and you should think about correcting it. The research is more focused on testing smaller samples and your statement about developing the moisturizing knit fabric is perhaps a bit ambiguous at this level.
Throughout the paper you repeat the same phrases, which is not appropriate or necessary. Please review and correct (some parts are highlighted in the text).
Please check the english style (language) of whole paper.
The characterization of microcapsules is not adequately explained and some analyzes of surface structure using reliable methods of tested samples are omitted.
You have not explained in section 2.2 that the model of skin particle detector used in the experiment is sample Cor- 98 neometer® CM825.
Also, the results in the tables could be improved for the readers, perhaps some better explanations could be used for the samples tested; it is not easy to follow the labels.
The discussion could be improved and conclusions need to be added.
The references are relevant to the research.
In general, the paper presents a good idea for research and could be a good basis for testing the efficacy on a model or textile sample with large proportions inthe development of a moisturizing knitted fabric.
After corrections and refreshing the paper, it could be published in the journal Materials.
Yours sincerely,
Reviewer

Author Response
please check

Reviewer 3 Report
1. The authors need to supplement the description of the research method: do you need to specify what fixed pressure was created on the surface of the human body from a textile part? Important: the same and constant normalized pressure of the testil on the skin surface is an important requirement for the experimental study of skin moisture.
2. The authors must present a preliminary calculation of the sample size when planning the experiment, taking into account the probability theory and taking into account the error of technical measurements (15 people need to be justified).
3. The exact conditions for removing textile parts from the surface of a person's hand are not clear from the article. How and with what devices is the same level of friction between textiles and leather ensured? After what exact time (seconds?) after removing textiles from the surface of a person's hand, is the skin moisture measured? These conditions should be included in the description of the research method.
4. If Table 2 and Figure 6 show the same quantitative data, then Figure 6 is not required in the article.
5. Fig.6 incorrectly presents analytical data, since the X-axis parameters of the samples have a different structure and do not have the unity of the physical parameter for the X-axis. Fig.6 is presented incorrectly, it must be removed from the article. Similar remarks to Fig.7. Figure 7 is presented incorrectly. The points for 15 samples of fundamentally different structures, which do not have a single system of physical measurements along the X axis, cannot be combined with a single curve on the graphs. Such graphical analysis is uncorrected. For the presented data in Fig.7, the table is sufficient. 5. Fig.7 needs to be deleted (it is incorrect).
6. The last line in the table.4. with "Significance", conclusions about the impact of this textile variant cannot be taken into account (0.062 is an insufficient level of stability of the solution).
7. The list of references does not take into account modern achievements on the topic of research received by the authors of scientific centers from Europe, America and many other countries. The authors of the article should finalize the studied literature and include new publications by authors from other countries in the literary review.
Author Response
please check

Round 2
Reviewer 1 Report
Good job. However, you still need to check the number of decimal places in Tables 3-5. Moreover, author indicated "...I hope ..." in conclusion, who am I in this case?
Author Response
Response to Reviewer
Thank you for your valuable comments
1.The number of decimal places in the table has been unified.
2."...I hope ..." in conclusion is changed to "it can be considered to ...".All the authors hope to do more in-depth research in the future
Thank you and best regards.
Reviewer 3 Report
The article can be published. However, the authors did not give an answer to question No. 2 and did not make appropriate changes to the article (about the necessary mathematical estimation of the sample size of the study based on probability theory). This remark and recommendation for the authors remains.
Author Response
Response to Reviewer
Thank you for your valuable comments
Combined with relevant literature [1],in the article, skin hydration and trans epidermal water loss are Interpreted variables,yarn blending ratio and structure are explanatory variables: k.
The minimum sample size is the lower limit of the sample size required to obtain the estimated parameter without considering its quality, i.e. n ≥ k+1, although parameter estimators can be obtained, n is too small to obtain high quality parameters. In addition, the sample size is too small to test the model.
Therefore, at least n ≥ 3 (k+1)=9 can meet the basic requirements of the model. In comprehensive consideration, the sample size is set at 15.
[1]Wegrzyn J E, Litzke W L, Gurevic M, et al. DOE/BNL Liquid natural gas heavy vehicle program[J]. SAE transactions, 1998: 831-836.
https://www.osti.gov/biblio/771104.
Thank you again for your comments and suggestions.